# Taming nitroformate through encapsulation with nitrogen-rich hydrogen-bonded organic frameworks

Jichuan Zhang[1,2,3], Yongan Feng [4], Richard J. Staples [5], Jiaheng Zhang[2,3✉] & Jean'ne M. Shreeve [1✉]

Owing to its simple preparation and high oxygen content, nitroformate [$^-C(NO_2)_3$, NF] is an extremely attractive oxidant component for propellants and explosives. However, the poor thermostability of NF-based derivatives has been an unconquerable barrier for more than 150 years, thus hindering its application. In this study, the first example of a nitrogen-rich hydrogen-bonded organic framework (HOF-NF) is designed and constructed through self-assembly in energetic materials, in which NF anions are trapped in pores of the resulting framework via the dual force of ionic and hydrogen bonds from the strengthened framework. These factors lead to the decomposition temperature of the resulting HOF-NF moiety being 200 °C, which exceeds the challenge of thermal stability over 180 °C for the first time among NF-based compounds. A large number of NF-based compounds with high stabilities and excellent properties can be designed and synthesized on the basis of this work.

[1] Department of Chemistry, University of Idaho, Moscow, ID, USA. [2] Research Centre of Flexible Printed Electronic Technology, Harbin Institute of Technology, Shenzhen, China. [3] Zhuhai Institute of Advanced Technology Chinese Academy of Sciences, Biomaterials Research Center, Zhuhai, China. [4] School of Environmental and Safety Engineering, North University of China, Taiyuan, China. [5] Department of Chemistry, Michigan State University, East Lansing, MI, USA. ✉email: zhangjiaheng@hit.edu.cn; jshreeve@uidaho.edu

An oxidizer is essential during the course of energy-releasing reactions in energetic materials (including propellants and explosives), which have a core role in defense and aerospace[1,2]. Among all reported oxidizer components, nitroformate [$^-C(NO_2)_3$, NF] is extremely attractive because (1) its high oxygen content, such as its classic derivative, hydrazinium nitroformate (HNF), with an oxygen balance of +13.11%, which is higher than that of most oxidizers[3,4]; (2) it always gives rise to high energy density compounds[5–7] which are attractive to energetic scientists; (3) when compared with the current oxidizer component, perchlorate, it avoids pollution caused by chlorine with almost zero pollution to the environment upon decomposition; and (4) when involved in a one-step synthesis which gives it a huge advantage over the vast majority of oxidizers which require complicated steps with concomitant high cost[3,4,8]. Since its first synthesis in 1850, NF has been widely applied in energetic materials[9]. Unfortunately, stability and high oxygen content are contradictory, especially for species that have several nitro groups, viz., nitroformate.

For more than 100 years, numerous attempts[5–7,10–17] to improve the stability of NF, for example by encapsulation in 3D MOFs or polymers with the aim to reduce the sensitivity to mechanical stimulation, were not successful. Almost all NF-based compounds decompose in the temperature range of 80–150 °C, thus not meeting the requirements of modern energetic materials for applications in thermostability (>180 °C)[18]. Notably, poor thermostability is the key limiting factor in its applications, and improving this characteristic seems to be an unconquerable barrier and a huge challenge.

Hydrogen-bonded organic frameworks (HOFs) are an emerging class of materials, which are constructed by self-assembly of organic building units through intermolecular hydrogen bonds (H-bonds) and have shown a greater potential in gas storage, chemical separation, and catalysis[19–21]. Contrary to MOFs and COFs (covalent organic frameworks), HOFs are easily prepared, their structure can be easily elucidated using single-crystal X-ray diffraction, the starting materials are readily found, and their regeneration processes are low energy-consuming[22,23]. In particular, HOFs show excellent thermal stability[24–28]. For example, HOF-19[23] and HOF-TCBP[24] show thermal stabilities as high as 350 and 395 °C (Fig. 1a, b), respectively. Furthermore, a large number of H-bonds between guest molecules or ions and the framework can enhance the thermal stability of HOFs. However, due to their high porosity, low density, and low oxygen balance, HOFs are commonly considered unsuitable for energetic materials applications. Interestingly, a large group of stable nitrogen-rich poly-amino and heterocyclic compounds such as poly-amino azoles and azines or fused rings have hydrogen bonding in their crystal structures[29–31]. This leads us to assume that these compounds can be incorporated into HOFs as energetic guest molecules.

Compared to other energetic compounds[5,6,32,33], NF with one negative charge and three $NO_2$ groups has the capacity to generate a large number of O···H bonds and strong electrostatic interactions such as ionic bonds with surrounding organic cations. O···H bonds often play an important role in stabilizing energetic materials. For example, some classic energetic compounds such as 1,1-diamino-2,2-dinitroethene (FOX-7), 2,4,6-triamino-5-nitropyrimidine-1,3-dioxide (ICM-102) and 1,3,5-triamino-2,4,6-trinitrobenzene (TATB) are highly thermally stable and insensitive resulting primarily from strong inter/intramolecular O···H bonds in their structures[34]. It is known that (1) poly-amino nitrogen-heterocyclic compounds readily form salts[5,29,35], (2) some cocrystals can contain neutral compounds and salts[36,37], and (3) oxygen atoms of oxygen-rich anions can be effectively trapped by H-bonds from surrounding ligands in energetic cocrystals and energetic salts. If poly-amino nitrogen-rich ligands are chosen appropriately to assemble with nitroformate, an energetic HOF connected by numerous coupled H-bonds is likely to form, with NF anions occupying the pores of the resulting HOF, surrounded by a large number of O···H bonds. As a result, the thermostability of the encapsulated NF is expected to improve markedly.

## Results

**Single-crystal structure.** Melaminium nitroformate (MaNF), and 3,6,7-triamino-7H-[1,2,4]triazolo[4,3-b][1,2,4]triazole (TATOT) were chosen to assemble a new energetic HOF-NF in the water at room temperature (Fig. 1c), where NF is the counter ion which occupies the pores of the resulting HOF. Single-crystal X-ray diffraction analysis shows that the HOF-NF crystallizes in the triclinic (P-1) space group with a density of 1.732 g cm$^{-3}$ at 100 K (Supplementary Table 1). Each unit cell contains one NF anion, one Ma cation, one neutral Ma molecule, and one neutral TATOT molecule. Among these species, the Ma cation and the neutral Ma and TATOT molecules were connected by a large number of coupled N···H bonds, forming an infinite plane (Fig. 2a, b) comprising two types of rings: six-membered rings (comprised of two neutral Ma molecules, two Ma cations, and two neutral TATOT molecules) and 8-membered rings (comprised of two Ma cations, four neutral Ma molecules, and two neutral TATOT molecules). The lengths of the N···H bonds in each layer range from 1.935 to 2.527 Å. The adjacent layers are also connected by two N···H bonds of 2.490 and 2.547 Å to form a 3D hydrogen-bonded organic framework with a pore diameter of ~6.20 Å (Fig. 2c). Between the two adjacent layers, a 6-membered ring and an 8-membered ring are alternating pattern (Supplementary Fig. 1), and the distance between adjacent layers is longer ~3.20 Å. This arises because the H-bonded hydrogen atoms stick out of the plane.

The NF anions are located between the two layers in two different arrangements: NF1 and NF2 (Fig. 3a). NF1 and NF2 were bonded with as many as 12 and 11 O···H bonds, respectively (Fig. 3b, c). NF1 is connected to 12 H-bonds, which is the largest number of H-bonds generated by nitroformates among all the reported NFs. The lengths of these 23 H-bonds fall between 2.150 and 2.568 Å. Although the diameter of NF is smaller than the pore diameter, due to the fact that the NF1 and NF2 are located symmetrically at the center of the pore (Supplementary Fig. 2), and a pair of nitro groups farther away from the center of the pore occupy the edge of the pore so that these nitro groups seem to be crowded within the pores, and the overall structure is similar to that of a cookie (framework) inlayed by nuts (NF) (Fig. 3d). It is interesting that every O atom of NF1 and NF2 forms at least one H-bond with the surrounding H atoms from the strengthened frameworks—this is the first example of an NF-based system where all the O atoms are H-bonded (Supplementary Figs. 3 and 4)[5–7,12–16]. Although the density of HOF-NF is lower than most modern energetic compounds[38–40], it has one of the highest densities reported for HOFs so far[19–28]. This is because (1) the pores, which are filled with NF1 and NF2, occupy about 17% of the volume of the unit cell, which is lower than most of the occupancy of pores among reported HOFs; and (2) the introduction of the highly dense NF. Notably, the interactions between ligands and the interactions between the framework and NF anions are enhanced by numerous multi H-bonds, which encourage the NF anions and framework to be a 3D network (Fig. 3a).

**Thermostability.** Sensitivity measurements show that HOF-NF is insensitive to impact and friction (IS > 40 J and FS > 360 N). The thermostability of HOF-NF was determined by DSC using a

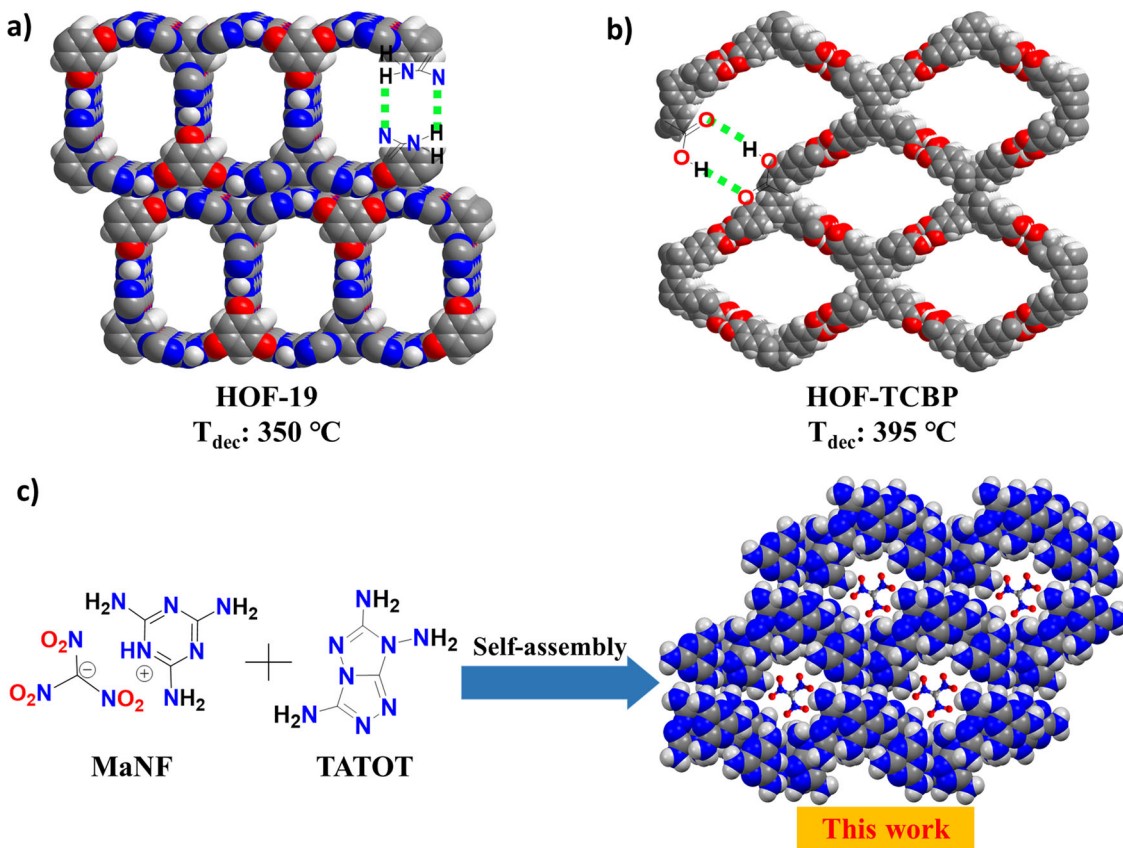

**Fig. 1 The structures of several HOFs. a** HOF-19 strengthened by coupled N⋯H bonds with large pores and high thermostability; **b** HOF-TCBP strengthened by coupled O⋯H bonds with large pores and high thermostability; **c** Self-assembly of HOF-NF by melaminiumnitroformate, and 3,6,7-triamino-7H-[1,2,4]triazolo[4,3-b][1,2,4]triazole (in this work).

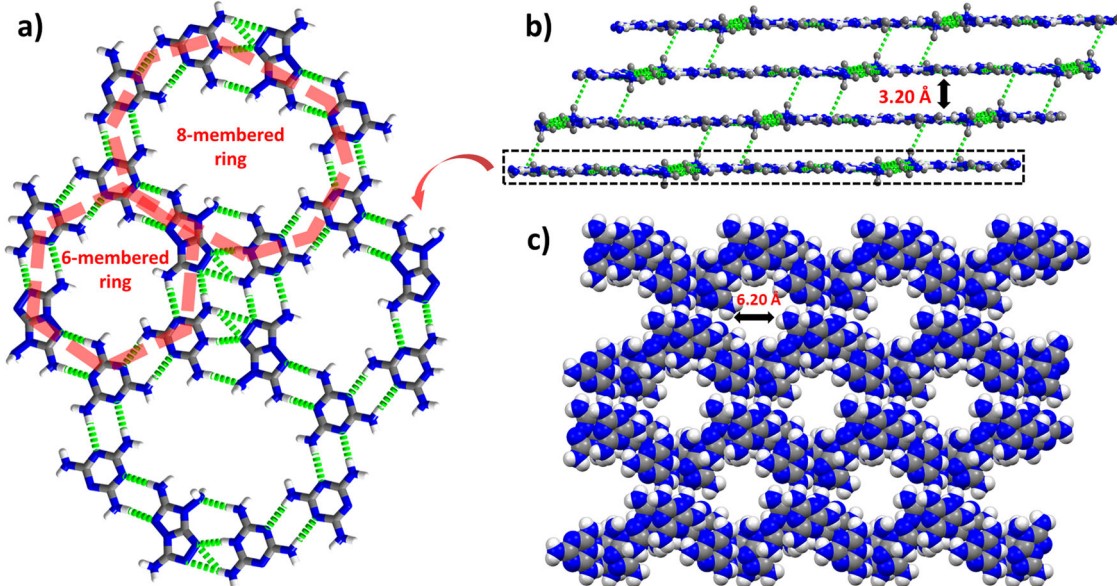

**Fig. 2 The packing mode of HOF-NF without NF anions. a** Arrangements of 6-membered rings and 8-membered rings in a single layer. **b** Layer–layer packing mode of the framework. **c** A packing diagram of framework showing the channel with a diameter of 6.2 Å. (Note: green dotted lines denoting the H-bonds, and NF anions are omitted here).

heating rate of 5 °C per minute under nitrogen atmosphere which indicates that the decomposition temperature (onset) is 200 °C (Fig. 4a and Supplementary Fig. 5). This is the first example of an NF- based system for which the decomposition temperature is >180 °C (Fig. 4b and Supplementary Table 2)[5–7,12–17,35,41,42]. The peak at 222 °C corresponds to the decomposition of NF, and the peak at 240.6 °C is attributed to the decomposition of TATOT[29]. Given its high thermostability, the Kissinger and

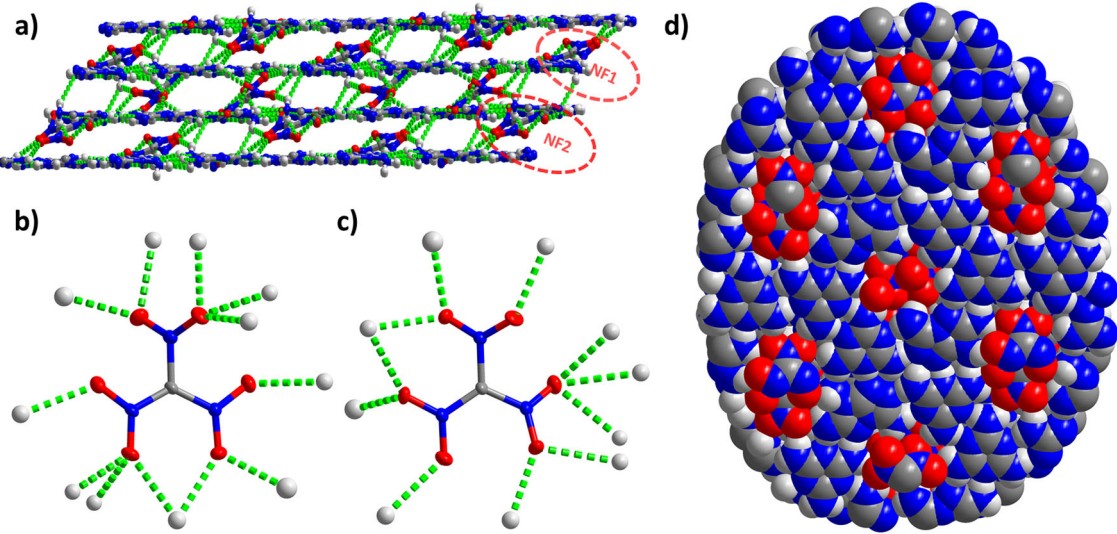

**Fig. 3 The packing mode of HOF-NF. a** 3D network of HOF-NF with two different arrangements of NF1 and NF2; **b, c** H-bonded environment of NF1 (left) and NF2 (right); **d** Overall structure view from *a* axis (space-filling mode).

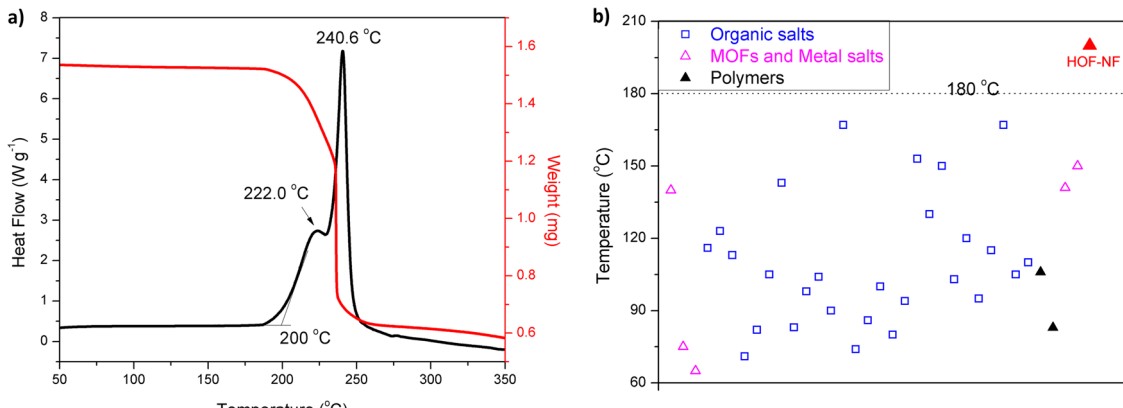

**Fig. 4 Thermostability of HOF-NF and the distribution of decomposition temperatures for all NF-based compounds. a** Thermal behavior (TG-DSC) of HOFs-NF; **b** The decomposition temperatures of reported NF-based compounds.

Ozawa methods[43] were employed to investigate the thermodynamic properties.

The DSC curves of HOF-NF at different heating rates (Fig. 5) show that the activation energies are 293.4 and 287.2 kJ mol$^{-1}$ (Supplementary Table 3), respectively. These values are not only considerably higher than those of classic explosives 2,4,6-trinitrotoluene (TNT ~115 kJ mol$^{-1}$)[44], cyclotrimethylenetrinitramine (RDX ~150 kJ mol$^{-1}$), but also higher than that of TATB (~211 kJ mol$^{-1}$)[45], which indicates that HOF-NF has excellent heat-resistant properties, and it is an excellent candidate as a steady and reliable explosive. The insensitivity, high thermostability and high heat-resistance of HOF-NF result from the electrostatic interaction between NF anions and the surrounding strengthened cationic frameworks and the fact that all O atoms of the NF anion are stabilized through H-bonding. The strengthened frameworks constructed by coupled N⋯H bonds were confirmed by NCI calculations (Fig. 6a)[46,47], in which most of the strong HB1 interactions were found on the ring. All O atoms of NF were surrounded by HB2 and HB3 interactions; extensive π–π interactions between two layers were also found. The strengthened frameworks were also confirmed by 2D fingerprint plots of Hirshfeld surfaces (Fig. 6b, c)[48], in which the percentage of N⋯H interactions (28.0%) is higher than the O⋯H interactions (24.5%). The strength of N⋯H interactions of HOF-NF is greater than that

of O⋯H interactions in almost all NF-based compounds (Supplementary Fig. 6). The C–N interactions (8.2%) represent the π–π interactions of the layer–layer interaction.

Diaminoguanidinium (DGNF, with 11 H bonds, Supplementary Fig. 4), and melamine nitroformates (MaNF with 10 H bonds) were chosen as representatives of NF-based compounds with many H-bonds. Formamidinium nitroformate (FNF, $T_{dec}$: 167 °C, Supplementary Table 2), and 4-amino-1,2,4-triazolate nitroformate (4ATNF, $T_{dec}$: 167 °C) were chosen as representatives of NF-based compounds with high thermostabilities, to calculate their electrostatic potentials (ESP)[49], and compared with HOF-NF (Supplementary Figs. 7–11). It was found that the average values of the maximum and minimum ESP around NF anions in these four compounds are considerably higher than the values in HOF-NF (Supplementary Table 3). The lower ESP indicates much greater structural stability[47,50]. The low ESP in HOF-NF is believed to be due to the strengthened framework and a large number of H-bonds which can stabilize NF anions very well. In summary, both calculation and experiment clearly confirm the insensitivity and high thermostability of HOF-NF.

**Energetic properties.** The heat of formation of HOF-NF was calculated according to the literature[32], and was determined to be

376.4 kJ mol$^{-1}$. Its detonation velocity and pressure were found to be 7817 m s$^{-1}$ and 21.63 GPa (Supplementary Tables 4 and 5), respectively, by EXPLO 5 (6.01 version)[51], using an experimental density of 1.68 g cm$^{-3}$ (room temperature). These detonation velocity and pressure values are higher than those of TNT[47]. Although the detonation performance of HOF-NF is lower than that of classical explosives such as RDX, the NF-based system is the first of its kind that meets the demand of modern energetic materials and it is also a potential candidate as an insensitive energetic material.

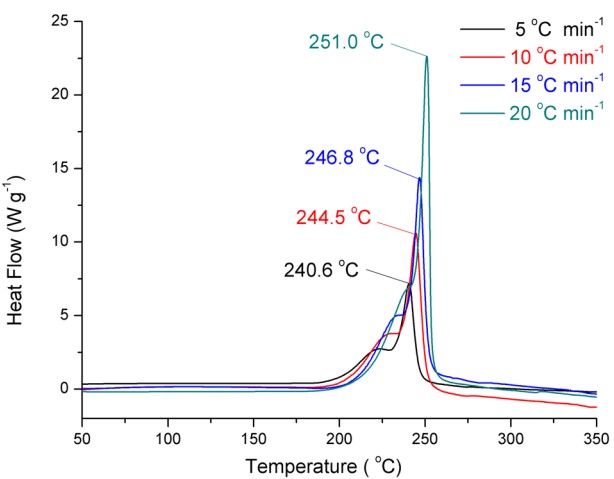

**Fig. 5 DSC curves of HOF-NF.** DSC curves of HOF-NF at different heating rates.

## Discussion

An energetic nitrogen-rich hydrogen-bonded organic framework (HOF-NF) was designed and obtained through self-assembly, whereby nitroformate anions occupy the cavities and are stabilized by a large number of H-bonds from the strengthened framework. Each of the oxygen atoms of NF participates in at least one H-bond, and these factors result in a thermostability of 200 °C, the highest thermostability among all NF-based compounds reported. In addition, the insensitivity and detonation performance support HOF-NF as the first example of an NF derivative with an actual potential application. We believe that this strategy will pave the way to stabilize nitroformate and to overcome the poor thermostability of nitroformate by encapsulation in HOFs. It is likely that an increasing number of HOFs, constructed of nitrogen-rich compounds that trap NF moieties, which exhibit excellent thermostability and higher detonation performance can be designed on the basis of this work.

## Methods

**Safety precautions.** Although none of the energetic compounds described herein have exploded or detonated in the course of this research accidentally, these materials should be handled with extreme care using the best safety practices.

**General.** Melamine was purchased from AKSci and was used as supplied. Melaminium nitroformate (MaNF) and 3,6,7-triamino-7H-[1,2,4]triazolo[4,3-b][1,2,4] triazole (TATOT) was synthesized according to the literature[5,29].$^1$H NMR and $^{13}$C NMR spectra were recorded on a 300 MHz (Bruker AVANCE 300) nuclear magnetic resonance spectrometer. Chemical shifts for $^1$H NMR and $^{13}$C NMR spectra are given with respect to external (CH$_3$)$_4$Si ($^1$H and $^{13}$C). [D$_6$] DMSO was used as a locking solvent unless otherwise stated. IR spectra were recorded using KBr pellets with an FT-IR spectrometer (Thermo Nicolet AVATAR 370). The density was determined at room temperature by employing a Micromeritics AccuPyc II 1340 gas pycnometer. Decomposition temperature (onset) was

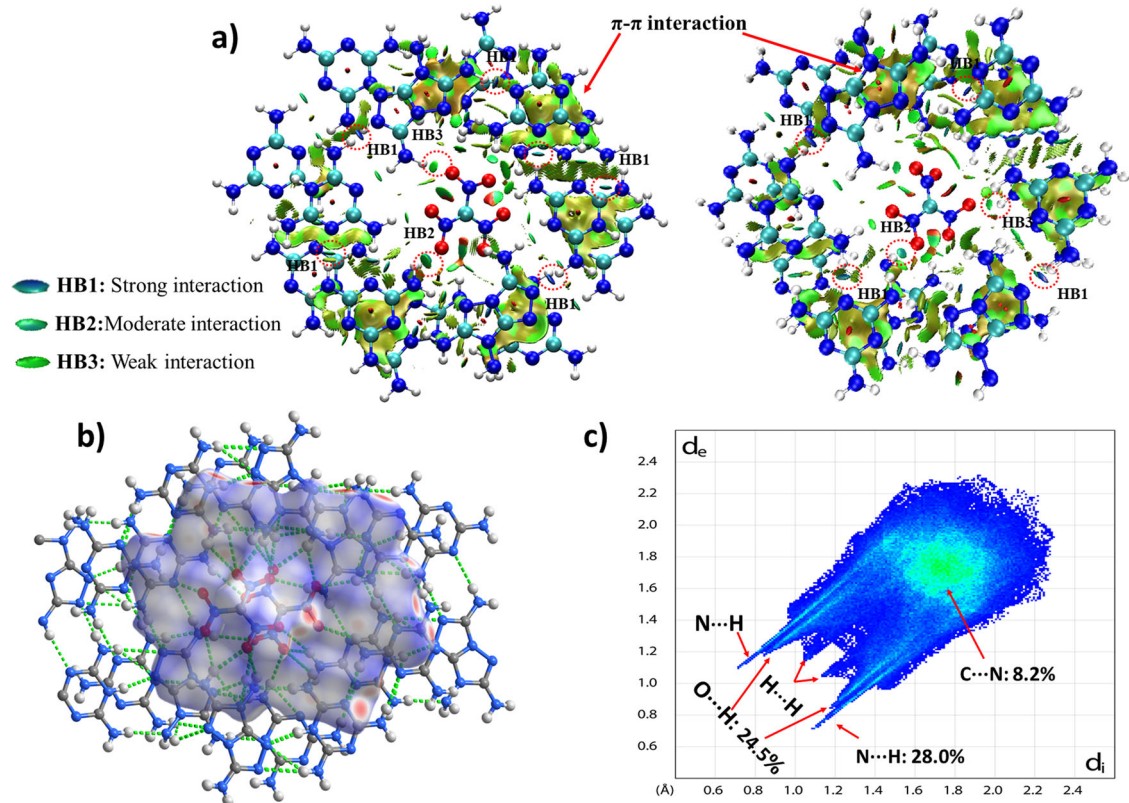

**Fig. 6 The weak interactions in HOF-NF. a** Noncovalent interaction (NCI) plots of NF1 (left) and NF2 (right) in HOF-NF, the surfaces are colored on a blue-green-red scale indicating strong attractive interactions, weak attractive, and strong nonbonded overlap, respectively; **b** Simulation of a Hirshfeld surface in a single ring; **c** 2D fingerprint plot generated by Hirshfeld surface.

recorded using dry nitrogen gas at a heating rate of 5 °C min$^{-1}$ and different rates (5, 10, 15, and 20 °C min$^{-1}$, respectively) on a differential scanning calorimeter (DSC, TA Instruments Q2000). Its thermo behavior was also recorded by a Thermogravimetric Analysis (TG, TAQ50). Elemental analyses (C, H, N) were performed with a Vario Micro cube Elementar Analyzer. Impact and friction sensitivity measurements were made using a standard BAM Fallhammer and a BAM friction tester.

**Synthesis of HOF-NF.** Melamine nitroformate salt (0.275 g, 1 mmol) and TATOT (0.157 g, 1 mmol) were added to water (30 mL), stirring at 60–70 °C for 30 min. The solution was filtered, and 2 or 3 days later, crystals formed at the bottom of the vial containing the filtrate. The crystals were obtained by filtration, washed with a small amount of water (5 mL), and dried.

**HOF-NF.** Orange crystal, yield: 75%. $^1$H NMR (d6-DMSO): δ 6.90 (s, 2H), 6.67 (s, 12H), 6.14 (s, 2H), 5.71 (s, 2H) ppm. $^{13}$C NMR (d6-DMSO): δ 165.3, 159.4, 148.2, 142.4 ppm. IR (KBr): v 3476, 3414, 3339, 3203, 1687, 1641, 1548, 1518, 1478, 1454, 1398, 1273, 1160, 994, 919, 865, 815, 785, 730, 671, 374, 496 cm$^{-1}$. Elemental analysis: $C_{10}H_{19}N_{23}O_6$ (557.48): Calcd: C 21.55, H 3.44, N 57.80%. Found: C 21.50, H 3.46, N 57.95%.

## Data availability

Data that support the finding of this study are available from the corresponding authors on request. X-ray coordinates from the crystal structure determination have been deposited with the Cambridge Crystallographic Data Centre (CCDC), under deposition number 2027045. These data can be obtained free of charge from The Cambridge Crystallographic Data Centre via www.ccdc.cam.ac.uk/data_request/cif. Some related original calculated input files and corresponding resulting files are provided and these data are available to the public.

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

## Acknowledgements

Financial support of the Office of Naval Research (N00014-16-1-2089), and the Defense Threat Reduction Agency (HDTRA 1-15-1-0028) is gratefully acknowledged. The Rigaku Synergy S Diffractomer was purchased with support from the National Science Foundation MRI program (1919565). This work was supported by the National Natural Science Foundation of China (21905069), the Shenzhen Science and Technology Innovation Committee (JCYJ20180507183907224, KQTD20170809110344233), Economic, Trade and Information Commission of Shenzhen Municipality through the Graphene Manufacture Innovation Center (201901161514), and Guangdong Province Covid-19 Pandemic Control Research Fund 2020KZDZX1220.

## Author contributions

Jic.Z., Jia.Z., and J.M.S. designed the study. Jic.Z. performed most of the reactions and measurements. Y.F. finished the theoretical calculations. R.J.S. performed the crystallographic structural analysis. Jic.Z., Jia.Z., and J.M.S. prepared the manuscript. Y.F. gave important suggestions for the paper.

## Competing interests

The authors declare no competing interests.
