## [Peer Review File · Nature Communications]

REVIEWERS' COMMENTS

Reviewer #1 (Remarks to the Author):

I previously reviewed this manuscript for [redacted] and recommended acceptance after major revision. Now a revised version has been sent by the authors to Nature Communications. In this present revised version the authors addressed to all my comments and made an already very good manuscript truly outstanding. Consequently, I recommend acceptance. No revision is necessary. Congratulations to the authors !

Reviewer #3 (Remarks to the Author):

No changes are made to the paper that address the main concern of significance. The minor details corrected do not change the basic fact that the material does not have exceptional properties nor is the approach particularly innovative and general. The authors stumble across a more thermally stable material than previously related materials and in doing so solved a problem that is not really considered a problem in the field (the authors argue this is a longstanding problem, but I think this convolutes the notion of a problem being longstanding with it being important). Even putting this aside, the material ultimately produced is far too low power for applications.

Reviewer #4 (Remarks to the Author):

Through careful reading of the author's manuscript and feedback on the reviewers' comments, I believe that the author has basically answered the questions raised by the reviewers seriously and agreed to accept the publication.

Reviewer 1:

Comments: I previously reviewed this manuscript for [redacted] and recommended acceptance after major revision. Now a revised version has been sent by the authors to Nature Communications. In this present revised version the authors addressed to all my comments and made an already very good manuscript truly outstanding. Consequently, I recommend acceptance. No revision is necessary. Congratulations to the authors !

Response: Thank you for your positive comments.

Reviewer 3:

Comments: No changes are made to the paper that address the main concern of significance. The minor details corrected do not change the basic fact that the material does not have exceptional properties nor is the approach particularly innovative and general. The authors stumble across a more thermally stable material than previously related materials and in doing so solved a problem that is not really considered a problem in the field (the authors argue this is a longstanding problem, but I think this convolutes the notion of a problem being longstanding with it being important). Even putting this aside, the material ultimately produced is far too low power for applications.

Response: We are sorry that you fail to be impressed by our work. Throughout the history of science and technology, mature technology and products always require continuous development and optimization, even if most of them are not promising at the beginning. Just like nitroformate, we all want the nitroformate-based compounds with high thermostabilities (higher than 200°C or even higher) and excellent detonation properties (comparable to those of HMX, Cl-20), but for present, it is less realistic, so patience is required. Although the resulting compound doesn't exhibit excellent energetic performance comparable to those high energy density materials (such as HMX), it solved issue of the poor thermostability of nitroformate. We believe that this work will enlighten scientists to develop much more nitroformate-based compounds with excellent properties (such as high decomposition temperature, high detonation performance), and also we will continue to design and prepare many more nitroformate-based compounds with high decomposition temperature and excellent detonation performance based on this work.

Reviewer 4

Comments: Through careful reading of the author's manuscript and feedback on the reviewers'

comments, I believe that the author has basically answered the questions raised by the reviewers seriously and agreed to accept the publication.

Response: Thank you for your positive comments.